# Early Serum Creatinine Levels after Aneurysmal Subarachnoid Hemorrhage Predict Functional Neurological Outcome after 6 Months

**DOI:** 10.3390/jcm11164753

**Published:** 2022-08-15

**Authors:** Tim Lampmann, Alexis Hadjiathanasiou, Harun Asoglu, Johannes Wach, Tamara Kern, Hartmut Vatter, Erdem Güresir

**Affiliations:** Department of Neurosurgery, University Hospital Bonn, 53127 Bonn, Germany

**Keywords:** subarachnoid hemorrhage, intracranial aneurysm, neurological outcome, serum creatinine, renal function, risk factor

## Abstract

Acute kidney injury (AKI) is a known predictor of unfavorable outcome in patients treated at the ICU, irrespective of the disease. However, data on the potential influence of serum creatinine (sCr) on hospital admission on the outcome in patients suffering from aneurysmal subarachnoid hemorrhage (SAH) is scarce. A total of 369 consecutive patients suffering from SAH were included in this retrospective cohort study. Patients were divided into good-grade (WFNS I–III) versus poor-grade (WFNS IV–V). Outcome was assessed according to the modified Rankin Scale (mRS) after 6 months and stratified into favorable (mRS 0–2) versus unfavorable (mRS 3–6). SAH patients with sCr levels <1.0 mg/dL achieved significantly a favorable outcome more often compared to patients with sCr levels ≥1.0 mg/dL (*p* = 0.003). In the multivariable analysis, higher levels of sCr (*p* = 0.014, OR 2.4; 95% CI 1.2–4.7), poor-grade on admission (*p* < 0.001, OR 9.8; 95% CI 5.6–17.2), age over 65 years (*p* < 0.001, OR 3.3; 95% CI 1.7–6.1), and delayed cerebral ischemia (*p* < 0.001, OR 7.9; 95% CI 3.7–17.1) were independently associated with an unfavorable outcome. We identified increased sCr on admission as a predictor for unfavorable functional outcome after SAH. Further studies elucidating the pathophysiology of this association are necessary.

## 1. Introduction

Patients suffering from aneurysmal subarachnoid hemorrhage (SAH) are at high risk for achieving an unfavorable neurological outcome, mainly due to the severity of the bleeding, delayed cerebral ischemia (DCI), and rebleeding [1,2]. Due to early treatment strategies, the risk of rebleeding has been reduced. The medical treatment of patients suffering from SAH has been substantially improved and medical complications, their prevention, and treatment present a cornerstone of the complex SAH treatment. While the focus of research was on DCI, up to now, no major therapeutic breakthrough affecting DCI has been achieved.

In critically ill patients as well as in SAH patients, acute kidney injury (AKI) is a frequent complication and its negative impact on the outcome is well-known [3]. In a large retrospective cohort study with over 5000 patients included, 16.3% of the patients who were admitted at the intensive care unit (ICU) because of a neurological disease reached class ‘risk’, 21.9% class ‘injury’, and 21.7% class ‘failure’ according to the RIFLE criteria [3,4]. A main criterion for the definition of AKI is an increase in serum creatinine (sCr) or urine output. These require repeated measurements and may be influenced by treatments already undertaken. Because of repeated measurements, an early prediction regarding the upcoming treatment course may be delayed or inappropriate. Data on sCr in patients suffering from SAH is scarce, especially the predictive value of early sCr on the outcome. We therefore performed an institutional analysis to describe the association of early sCr and the neurological outcome.

## 2. Materials and Methods

### 2.1. Patient Population and Clinical Management

A total of 371 consecutive patients suffering from SAH were treated between January 2011 and December 2016 at the authors’ institution. Early sCr was available for 369 patients on admission and those were included in further analysis. The SAH was verified by computed tomography (CT) or lumbar puncture. Patient characteristics, treatment modality, aneurysm size and location, laboratory measurements, radiological features, and functional neurological outcome were assessed and entered into a computerized database (SPSS, Version 27, IBM Corp., Armonk, NY, USA).

We followed an early treatment strategy in all clinical SAH grades within 48 h after admission [5,6]. Within 24 h after aneurysm treatment, all patients received a CT scan to identify possible periprocedural complications, thus better differentiating later DCI during treatment course. Every patient included in the analysis underwent laboratory examinations on admission before any diagnostic measures or treatment. All patients were monitored on the neurosurgical ICU and received medical treatment with nimodipine. In cases of DCI or delayed ischemic neurological deficit (DIND), induced hypertension with catecholamines was initiated [7]. Patients presenting with additional subdural hematoma or aneurysm related intracerebral hemorrhage were treated surgically [8,9]. In cases of refractory elevated intracranial pressure, decompressive hemicraniectomy was performed as previously described [10].

Patients were graded with the World Federation of Neurosurgical Societies (WFNS) system according to their presenting neurological status on admission and were stratified into good-grade (WFNS I–III) versus poor-grade (WFNS IV–V). Outcome was assessed according to the modified Rankin Scale (mRS) after 6 months and stratified into favorable (mRS 0–2) versus unfavorable (mRS 3–6).

### 2.2. Laboratory Examinations 

The laboratory information system Lauris (version 17.06.21, Swisslab GmbH, Berlin, Germany) was used for retrospective data acquisition. Blood samples are taken routinely on admission to the emergency room before any diagnostic or therapeutic measures. The sCr was determined by VIS-photometry using the Jaffe reaction with a CREJ2 reagent (Roche, Basel, Switzerland).

The cut-off for sCr on admission was set at 1 mg/dL according to the literature [11]. Patients were stratified into low sCr (sCr < 1 mg/dL) and high sCr (sCr ≥ 1 mg/dL) on admission. The medical records and laboratory results of patients with high sCr were additionally examined prior to SAH to further investigate whether AKI was pre-existent to SAH. 

### 2.3. Statistical Analysis

The data analysis was executed using SPSS Statistics (Version 27, IBM Corp. Armok, NY, USA). The unpaired *t*-test was used for parametric statistics. Categorical variables were analyzed in contingency tables using the Fisher exact test. Results with *p* < 0.05 were considered statistically significant. Furthermore, a multivariable analysis was performed to find the independent predictors of an unfavorable functional outcome in patients suffering from SAH using binary logistic regression analysis to find confounding factors between potentially independent predictors. Variables with significant *p*-values in the univariate analysis or known variables influencing the neurological outcome in SAH were considered as potentially independent variables in this multivariable analysis. A backward stepwise method was used to construct a multivariable logistic regression model in relation to an unfavorable outcome as a dependent variable with an inclusion criterion of *p* < 0.05.

## 3. Results

The information on patient characteristics are shown in Table 1. The levels of serum creatinine regarding the achieved outcome are shown in Figure 1.

### 3.1. Univariate Analysis

A total of 297 (80%) patients had low sCr and 72 (20%) patients had high sCr on admission. A favorable outcome was achieved more often by patients with low sCr on admission compared to patients with high sCr (*p* < 0.001) in this cohort. The details are shown in Table 2.

A total of 346 (94%) of the 369 patients received further treatment. Twenty-three (6%) patients were in such poor neurological condition not justifying further treatment or treatment was in contrast to the patients’ will. Looking only at patients that received treatment, those with low sCr on admission presented with lower WFNS grade and higher Glasgow Coma Scale (GCS) compared to patients with high sCr on admission (*p* = 0.03, respectively *p* = 0.038). Mean sCr in patients with low sCr was 0.71 ± 0.15 mg/dL and 1.5 ± 1.03 mg/dL in patients with high sCr (*p* < 0.001). Patients with low sCr were younger compared with patients with high sCr on admission (55 ± 13 vs. 60 ± 15, *p* = 0.003). The rate of clinically relevant CVS or DCI did not differ significantly between both groups. Favorable outcome was achieved more often by patients with low sCr on admission compared to patients with high sCr (*p* = 0.003). Further details are shown in Table 3.

### 3.2. Renal Function Prior to SAH

Information on renal function prior to SAH was available in 49 (68%) patients who presented with high sCr on admission. A comparison of sCr before the occurrence of SAH with sCr on admission in this sample showed no significant differences in the level of sCr.

### 3.3. Multivariable Analysis

We performed a multivariable logistic regression analysis of those variables significantly associated with an unfavorable outcome and known predictors of an unfavorable outcome. According to multivariable analysis, ‘high sCr’ (*p* = 0.014, OR 2.4; 95% CI 1.2–4.7), ‘poor-grade SAH’ (*p* < 0.001, OR 9.8; 95% CI 5.6–17.2), ‘age > 65 years’ (*p* < 0.001, OR 3.3; 95% CI 1.7–6.1), and ‘DCI’ (*p* < 0.001, OR 7.9; 95% CI 3.7–17.1) were independently associated with an unfavorable outcome (Nagelkerke’s R^2^ 43.2%; Figure 2).

## 4. Discussion

Due to advancements in neurocritical care, the research focus is shifting to non-neurological complications affecting the outcome in SAH. Many studies have reported the affection of extracerebral organs due to SAH such as the lungs, heart, liver, and kidney, while most of them have focused on cardiopulmonary complications [12,13,14]. AKI is a known predictor of an unfavorable outcome among patients treated at the ICU, independent of the primary disease [3]. Previous studies found an incidence of AKI in patients suffering from SAH in up to 25% [13,15,16,17]. It is therefore necessary to identify early predictors for AKI that may also predict the neurological outcome. This consequently distinguishes these patients at higher risk for deterioration.

The main definition criteria for AKI are based on changes in the sCr levels compared to the baseline sCr or the urine output over time [4,18,19,20,21]. In this study, we investigated the initial sCr on admission prior to any diagnostic or therapeutical measures. The sCr values reported in this study do not dynamically represent the renal function, but indicate the renal function on admission after SAH that is not confounded due to an already conducted treatment with a readily available laboratory marker. The aim of this study was to determine whether elevated sCr on admission represents a risk factor for achieving an unfavorable outcome after SAH. 

In our series of 369 consecutive patients, we found that elevated sCr on admission independently predicted an unfavorable outcome after 6 months. Even after the exclusion of patients with extremely poor neurological condition on admission not justifying further treatment, sCr was still independently associated with an unfavorable outcome. Previous studies reported that even marginal increases in the sCr levels in ICU patients relevantly worsened the outcome [22]. Investigations that used the RIFLE and AKIN guidelines for AKI also proved that even moderate AKI raised the mortality risk [17,23]. However, Tujjar et al. and Eagles et al. reported that AKI did not independently influence the outcome [16,23]. Eagles et al. could not elucidate why AKI was independently associated with mortality, but not with functional outcome [23]. The relationship between AKI and mortality has some possible explanations. AKI occurs in about 57% of critically ill ICU patients and mortality is independently increased in patients suffering from AKI [24,25]. Underlying pathomechanisms remain unclear, but most notably, complications following AKI such as metabolic acidosis and cumulative fluid balance may cause mortality rates up to 16.2% [25]. Renal replacement therapy, which improves renal function, improves survival [25]. This strengthens the thesis that impaired renal function and its complications are responsible for deterioration. Our results could reflect the possibility of a reliable and readily available marker on admission that can predict a worse neurological outcome and could raise the awareness of the treating physicians in identifying patients at higher risk for neurological deterioration during an upcoming treatment course. A possible advantage of sCr on admission in contrast to the RIFLE and AKIN guidelines in this setting could be the presentation of renal function unaffected from treatment associated factors such as contrast agents.

In the present study, patients admitted with high sCr were older, more frequently of the male sex, and presented with lower GCS and, respectively, with poorer SAH grade. According to the multivariable analysis, anything but sex was an independent factor for an unfavorable outcome. This legitimizes the static cut-off of sCr of 1 mg/dL, which was not adjusted by sex. The available sCr data of patients presenting with high sCr before SAH were analyzed and did not show a significant difference. This leads to the assumption that elevated sCr on admission represents the basal renal function of these patients and not an immediate SAH-related elevation of sCr. According to these findings, elevated sCr can be interpreted as a predictive factor for an unfavorable neurological outcome due to a restricted renal function prior to SAH. Eagles et al. reported in an analysis of the data set of the CONSIOUS-1 trial that sCr on admission was not a risk factor for developing AKI during the treatment course of SAH, but AKI as a complication of SAH increased the probability of death [23]. Zacharia et al. also reported that even slight decreases in the creatinine clearance reduced the neurological outcome [17]. In the same study, pre-existing renal disease was not associated with the risk of developing AKI during the treatment course, but interestingly, lower sCr on admission was. Furthermore, pre-existing renal disease was not associated with an unfavorable outcome, but in accordance with our results, elevated sCr on admission was.

Previous reports investigated the influence of AKI on the outcome in SAH patients using the RIFLE and AKIN guidelines [17,23]. Terao et al. detected microalbuminuria over the first 8 days after SAH to be associated with a poor neurological outcome [26]. However, repeated measurements may be too intricate compared to the advantage of a one-time measurement of sCr on admission, allowing for the identification of patients at higher risk for deterioration as early as possible.

The sCr is already part of a decision tree algorithm as a predictor of mortality in patients suffering from TBI with traumatic subarachnoid hemorrhage [27]. In that study, a cut-off of 1.4 mg/dL was determined and elevated sCr predicted mortality in combination with age and head Abbreviated Injury Scale. In our series of SAH patients, an even lower cut-off was shown to be significant. Another study emphasized the urea–creatinine ratio (UCR) to be predictive for a poor clinical outcome following SAH [28]. Only patients with normal renal function were investigated and UCR was determined in the early (day 0–2) and critical (day 5–7) phase after surgery. The greatest advantage of an early determination of sCr with a cut-off of 1.0 mg/dL is a distribution at an early stage, even before renal function may be impaired due to treatment course.

### Limitations

Although our results suggest that initial high sCr predicts an unfavorable outcome, there are several limitations to our study. Data collection and analysis were conducted in a retrospective manner and represent a single center experience. Information of renal function prior to the SAH of patients presenting with high sCr on admission was not entirely available. However, due to the younger age of patients suffering from SAH than other stroke types, it is unlikely to expect a greater proportion of patients with pre-existing renal disease. Furthermore, other possible variables influencing sCr could not be acquired in this setting.

## 5. Conclusions

High sCr on admission was identified as an independent predictor for an unfavorable outcome after SAH. As a readily available marker, sCr allows physicians to identify patients straight on admission to be at higher risk for deterioration during an upcoming treatment course.

## Figures and Tables

**Figure 1 jcm-11-04753-f001:**
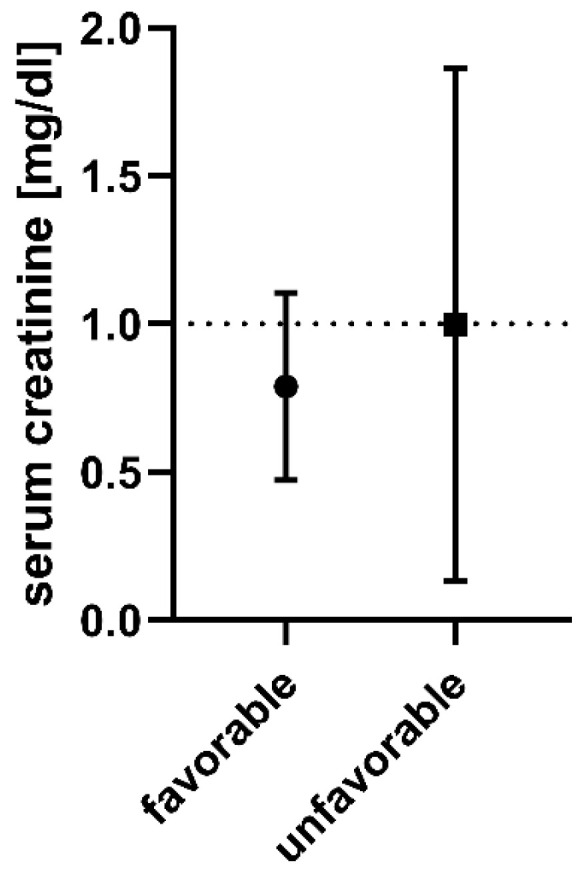
The means (black circle and square, respectively) ± SD of serum creatinine on the admission of patients that achieved favorable (modified Rankin scale 0–2) or unfavorable outcomes (modified Rankin scale 3–6). Dotted line at 1 mg/dL.

**Figure 2 jcm-11-04753-f002:**
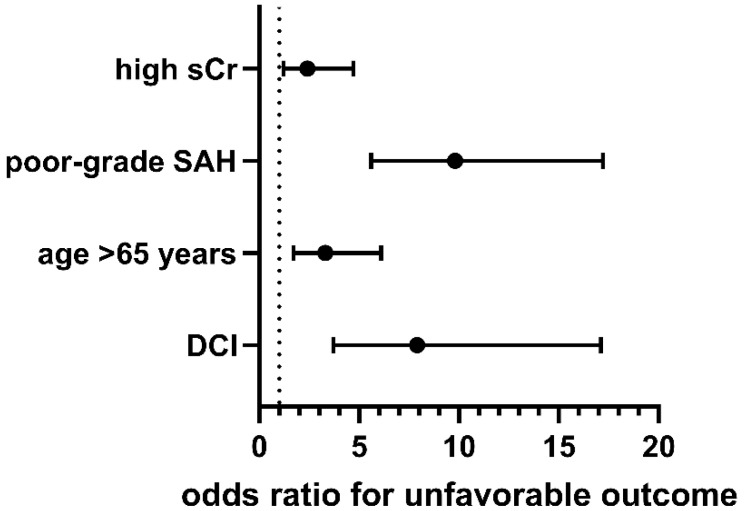
The forest plot presenting the results of the multivariable logistic regression analysis for an unfavorable outcome (modified Rankin scale 3–6) within the study group of patients suffering from SAH. Black circles represent odds ratio.

**Table 1 jcm-11-04753-t001:** The baseline characteristics of the SAH patients.

Characteristics	*n* = 369	(%)
**Mean age (±SD) [in years]**	56	±13.6
**Sex (Female)**	246	(67)
**Current smoking**	155	(42)
**Hypertension**	153	(42)
**Mean GCS (±SD)**	10.4	± 5
**Poor-grade SAH (WFNS grade IV + V)**	151	(41)
**Mean creatinine on admission (±SD) [in mg/dL]**	0.89	±0.67
**Early hydrocephalus**	248	(67)
**Aneurysm location**		
Acom/ACA	152	(41)
ICA	86	(23)
MCA	73	(20)
Posterior circulation	58	(16)
**Mean aneurysm size (±SD) [in mm]**	7.7	±5
**Treatment**		
Surgical	151	(41)
Endovascular	195	(53)
No treatment	23	(6)
**Vasospasm**	184	(50)
**DCI**	56	(15)
**Mean length of ICU stay (±SD) [in days]**	16.6	±15.4
**Favorable Outcome (mRS ≤ 2)**	185	(50)

ACA—anterior cerebral artery; Acom—anterior communicating artery; DCI—delayed cerebral ischemia; GCS—Glasgow Coma Scale; ICA—internal carotid artery; ICU—intensive care unit; mRS—modified Rankin scale; sCr—serum creatinine; SD—standard deviation; WFNS—World Federation of Neurosurgical Societies.

**Table 2 jcm-11-04753-t002:** The patient stratification according to low sCr and high sCr.

	Low sCr		High sCr		*p*-Value
	*n* = 297	(%)	*n* = 72	(%)	
**Mean age (± SD) [in years]**	55	±13	60	±15	**0.005**
**Sex (Female)**	220	(74)	26	(36)	**<0.001**
**Current smoking**	124	(42)	31	(44)	0.77
**Hypertension**	116	(39)	37	(51)	0.06
**Mean GCS (±SD)**	10.7	±4.9	8.8	±5.4	**0.003**
**Poor-grade SAH (WFNS grade IV + V)**	111	(37)	40	(56)	**0.007**
**Mean creatinine on admission (± SD) [in mg/dL]**	0.72	±0.15	1.61	±1.25	<0.001
**Early hydrocephalus**	198	(67)	50	(69)	0.68
**Aneurysm location**					0.24
Acom/ACA	120	(40)	32	(44)	
ICA	70	(24)	16	(22)	
MCA	64	(22)	9	(13)	
Posterior circulation	43	(14)	15	(21)	
**Mean aneurysm size (± SD) [in mm]**	7.6	±4.9	8	±5.3	0.59
**Treatment**					**0.017**
Surgical	126	(35)	25	(42)	
Endovascular	158	(53)	37	(51)	
No treatment	13	(4)	10	(14)	
**Vasospasm**	156	(53)	28	(39)	**0.048**
**DCI**	51	(17)	5	(7)	**0.029**
**Mean length of ICU stay (± SD) [in days]**	17	±16	14.8	±12.7	0.29
**Favorable Outcome (mRS ≤ 2)**	163	(55)	22	(31)	**<0.001**

ACA—anterior cerebral artery; Acom—anterior communicating artery; DCI—delayed cerebral ischemia; GCS—Glasgow Coma Scale; ICA—internal carotid artery; ICU—intensive care unit; mRS—modified Rankin scale; sCr—serum creatinine; SD—standard deviation; WFNS—World Federation of Neurosurgical Societies. Significant *p*-Values (<0.05) are bold.

**Table 3 jcm-11-04753-t003:** A comparison of only the treated patients after stratification into low sCr and high sCr.

	Low sCr		High sCr		*p*-Value
	*n* = 284	(%)	*n* = 62	(%)	
**Mean age (±SD) [in years]**	55	±13	60	±15	**0.003**
**Sex (Female)**	209	(74)	23	(37)	**<0.001**
**Current smoking**	120	(42)	26	(42)	1
**Hypertension**	112	(39)	32	(51)	0.09
**Mean GCS (± SD)**	11	±5.2	9.5	±5.3	**0.038**
**Poor-grade SAH (WFNS grade IV + V)**	99	(35)	31	(50)	**0.03**
**Mean creatinine on admission (±SD) [in mg/dL]**	0.71	±0.15	1.5	±1.03	**<0.001**
**Early hydrocephalus**	188	(66)	44	(71)	0.55
**Aneurysm location**					0.2
Acom/ACA	117	(41)	30	(48)	
ICA	70	(25)	13	(21)	
MCA	59	(21)	7	(11)	
Posterior circulation	13	(13)	12	(20)	
**Mean aneurysm size (±SD) [in mm]**	7.5	±4.8	7.9	±5.6	0.54
**Treatment**					0.58
Surgical	126	(44)	25	(40)	
Endovascular	158	(56)	37	(60)	
**Vasospasm**	153	(54)	28	(45)	0.26
**DCI**	51	(18)	5	(8)	0.06
**Mean length of ICU stay (± SD) [in days]**	17.4	±16.1	16.7	±12.7	0.76
**Favorable Outcome (mRS ≤ 2)**	162	(57)	22	(36)	**0.003**

ACA—anterior cerebral artery; Acom—anterior communicating artery; DCI—delayed cerebral ischemia; GCS—Glasgow Coma Scale; ICA—internal carotid artery; ICU—intensive care unit; mRS—modified Rankin scale; sCr—serum creatinine; SD—standard deviation; WFNS—World Federation of Neurosurgical Societies. Significant *p*-Values (<0.05) are bold.

## Data Availability

The raw data of this study are available from the corresponding author upon reasonable request.

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
