# Peer review of "Early Serum Creatinine Levels after Aneurysmal Subarachnoid Hemorrhage Predict Functional Neurological Outcome after 6 Months"

_jcm, 2022, doi:10.3390/jcm11164753_

Round 1
Reviewer 1 Report
This manuscript addresses a creatinine level on admission as a prognostic factor for subarachnoid hemorrhage.
The result part and introduction and discussion part must be improved with more explanations and discussions.
Acute kidney injury (AKI) occurs as a complication of various severe systemic disease, whereas the creatinine level on admission evaluates complications at the onset of subarachnoid hemorrhage (SAH).
Author Response
Thank you for the useful comments. We believe that after incorporating the issues listed below, the manuscript is clearer and the study is now better understood. We hope that the manuscript now is eligible for publication.
We extended the introduction to outline current of AKI definition and their pitfalls. Page 1f:
“A main criterion for definition of AKI is increase in serum creatinine (sCr) or urine output. These require repeated measurements and may be influenced by already undertaken treatment. Because of repeated measurements, an early prediction regarding the upcoming treatment course may be delayed or inappropriate.”
We added a new figure in the results part to demonstrate the mean sCr in relation to outcome. See page 4.
We revised the discussion and discussed sCr in other diseases or other utilization of sCR in SAH. Page 9f:
“The sCr is already part of a decision tree algorithm as a predictor of mortality in pa-tients suffering from TBI with traumatic subarachnoid hemorrhage [27]. In that study a cut-off of 1.4 mg/dl was determined and elevated sCr predicted mortality in combination with age and head Abbreviated Injury Scale. In our series of SAH patients an even lower cut-off showed to be significant. Another study emphasizes urea–creatinine ratio (UCR) to be predictive for a poor clinical outcome following SAH [28]. Only patients with normal renal function were investigated and UCR was determined in an early (day 0-2) and criti-cal (day 5-7) phase after surgery. Most advantage of an early determination of sCr with a cut-off of 1.0 mg/dl is a distribution at an early stage, even before renal function may be impaired due to treatment course.”
Moreover, pathomechanisms of AKI causing higher rates of mortality were discussed, too. Page 8:
“The relationship between AKI and mortality has some possible explanations. AKI occurs in about 57% of critically ill ICU patients and mortality is independently increased in pa-tients suffering from AKI [24,25]. Underlying pathomechanisms remain unclear, but most notably complications following AKI like metabolic acidosis and cumulative fluid balance may cause mortality rates up to 16.2% [25]. Renal replacement therapy which improves renal function improves survival [25]. This strengthens the thesis that impaired renal function and its complications are responsible for deterioration.”
Reviewer 2 Report
The authors demonstrated increased sCr on admission as a statistically significant, independent predictor for unfavorable functional outcome after SAH. The result is informative and reasonable. I have no major criticism for revision, But here a small comment. As the authors mentioned in the manuscript, some authors could not find AKI did not independently influence the outcome. If the authors can assume the reason, it would be better to demonstrate in the manuscript. Second, why the AKI can cause the poor outcome after SAH. This pathophysiological mechanism could be discussed as well.
Author Response
Thank you for your pleasant review. We believe that after incorporating the issues listed below, the manuscript is clearer and the study is now better understood. We hope that the manuscript now is eligible for publication.
Tujjar et. al. found that AKI did not independently influence the outcome, but following independent predictors did: WFNS scale, Fisher scale, seizure and vasopressor therapy. We now discuss possible pathomechanisms of AKI causing higher rates of mortality. Page 9:
“The relationship between AKI and mortality has some possible explanations. AKI occurs in about 57% of critically ill ICU patients and mortality is independently increased in patients suffering from AKI [24,25]. Underlying pathomechanisms remain unclear, but most notably complications following AKI like metabolic acidosis and cumulative fluid balance may cause mortality rates up to 16.2% [25]. Renal replacement therapy which improves renal function improves survival [25]. This strengthens the thesis that impaired renal function and its complications are responsible for deterioration.”
Tujjar et. al. wrote “higher incidence of poor outcome in patients with AKI after SAH reflects the severity of the disease and of secondary medical complications in this setting” what aligns with our thesis. Most important statement of our study is that high sCr predicts unfavorable outcome even before renal function may be impaired or rather AKI occurred.
Round 2
Reviewer 1 Report
The manuscript has been much improved and is in a nice condition now.